# Lanthanum modulated reaction pacemakers on a single catalytic nanoparticle

Maximilian Raab [1], Johannes Zeininger [1], Yuri Suchorski [1], Alexander Genest [1], Carla Weigl [1] & Günther Rupprechter [1]✉

Promoters are important in catalysis, but the atomistic details of their function and particularly their role in reaction instabilities such as kinetic phase transitions and oscillations are often unknown. Employing hydrogen oxidation as probe reaction, a Rh nanotip for mimicking a single Rh nanoparticle and field electron microscopy for in situ monitoring, we demonstrate a La-mediated local catalytic effect. The oscillatory mode of the reaction provides a tool for studying the interplay between different types of reaction pacemakers, i.e., specific local surface atomic configurations that initiate kinetic transitions. The presence of La shifts the bistable reaction states, changes the oscillation pattern and deactivates one of two pacemaker types for the La-free surface. The observed effects originate from the La-enhanced oxygen activation on the catalyst. The experimental observations are corroborated by micro-kinetic model simulations comprising a system of 25 coupled oscillators.

The activity and selectivity of technological catalysts can be optimised by addition of small amounts of substances, which are not catalysts by themselves but improve the catalytic performance: promoters. Often, alkali metals are employed as promoters in heterogeneous catalysis, which motivated numerous model studies of alkali adsorption and coadsorption on single crystal surfaces by surface sensitive methods under ultrahigh vacuum (UHV)[1–8]. Due to experimental difficulties, studies of the promotor function at work, i.e., in situ during catalytic reactions, are less numerous and mainly focused on the activity and selectivity of the promoted catalysts in the active steady state[9–12]. Although processes out of the steady state, i.e., catalytic instabilities such as catalytic light-off or self-sustaining oscillations, may also be influenced by promoters, the corresponding effects were hardly considered until now.

Several phenomena have been identified to contribute to the alkali promotion effect: (i) electrostatic interaction resulting from electron donation[13,14], (ii) direct bonding between alkalis and adsorbates[15], (iii) alkali-induced electronic polarisation[16], and (iv) structural effects[17,18]. Since the promoting effect of coadsorbed alkalis stems mainly from electron donation, other electropositive elements (alkaline earth, rare earth) may have a similar effect as alkali in many, e.g., oxidation reactions[19,20], while still possessing generally higher thermostability than adsorbed alkalis[21]. However, they were much less studied in terms of their promoting action.

At usual reaction temperatures, the mobility of promoters convoluted with the mobility of reactants results in a complex two-dimensional evolution of surface concentrations of the species participating in the reaction. When subsurface species are also involved, this is extended to a third dimension. Such intricate behaviour is no longer accessible by ensemble-averaging techniques, which calls for spatially resolved methods that can monitor processes in situ and in real time. In case of catalytic reactions at pressures beyond UHV conditions, this creates a huge challenge for experimental investigations. Nevertheless, using photons[22], electrons[23–27], ions[25,27–29], or metastable He atoms[30], which all are directly emitted from the studied surface, for microscopic imaging, "watching" ongoing catalytic reactions in situ becomes possible. In the present study we use a combination of field ion microscopy (FIM) for the atomically resolved characterisation of the catalyst and field electron microscopy (FEM) for in situ imaging of the lanthanum modified catalytic $H_2$ oxidation on a single Rh catalytic particle, which is modelled by the apex of a Rh nanotip. This approach already allowed revealing several new effects in the "non-promoted" hydrogen oxidation reaction, such as multifrequential oscillations[31], local intraparticle pacemakers[32], transitions between different reaction

[1]Institute of Materials Chemistry, TU Wien, Getreidemarkt 9, 1060 Vienna, Austria. ✉e-mail: guenther.rupprechter@tuwien.ac.at

modes modulated by spatial coupling[33] and even emergence of chaos in a nanosized reaction system[34]. Modifying the reaction system in an atomically controlled way by addition of a submonolayer of an electropositive promotor enabled us to detect another novel effect in the present study: a lanthanum-modulated switching between different local surface atomic configurations acting as reaction pacemakers in $H_2$ oxidation on Rh.

## Results

### Sample characterisation and La deposition on a Rh nanoparticle

A Rh nanotip to be used as a model of a single catalytic particle was fabricated by electrochemical etching of a Rh wire and characterised by FIM. This technique visualizes the surface using field-ionised noble gas atoms (Ne in the present case) for the formation of a magnified point projection image (Fig. 1a), providing the positions of protruding surface atoms with atomic resolution[35] (Fig. 1b). FEM forms an analogue projection image but via electrons field emitted from the same surface, and thus mapping the local work function distribution[36] (Fig. 1c). The complementary FIM/FEM images of the clean sample allow determining the apex shape, dimensions, and crystallographic orientations of the nanofacets on the tip[32,34].

The hemispherical Rh tip apex was characterised by FIM with atomic resolution and its [001] orientation is evident from the 4-fold symmetry of the FIM image (Fig. 1b). The (hkl)-orientation of nanofacets on the apex was determined by comparison of the [001] stereographic projection of the fcc lattice with the nanofacet positions in the FIM image, based on symmetry arguments. Using the ring counting method[35] and a comparison to ball models representing the atomic structure of the sample, the radius of the hemispherical apex was determined to be 18 nm. The atomically clean Rh surface was subsequently imaged by FEM and regions of interest (ROIs), each $2 \times 2$ nm$^2$,

on the (001), (011) and (111) facets were chosen for the kinetic studies (Fig. 1c).

As adsorbates usually influence the work function of a metal surface, FEM can also be used to observe hydrogen[37], oxygen[38] or lanthanum adsorption on the Rh surface. Figure 1d shows an FEM image of the Rh apex surface covered with a sub-monolayer of La, in which a clear change in the pattern (cf. Fig. 1c) can be observed. The La-coverage was controlled via the directly measured work function which on transition metals linearly decreases up to a La-coverage of 0.5 ML and consequently gradually approaches a plateau upon achieving monolayer coverage[39–41] (see Supplementary Note 1 and Supplementary Fig. 1 for details). The strong repulsive interaction between La-adatoms, characteristic for submonolayers of electropositive adsorbates, provides a homogeneous La-coverage on the Rh surface[42].

### Kinetic transitions and bistability

For catalytic experiments, the setup was used in the FEM mode and the FIM/FEM chamber was utilised as a flow reactor, exposing the Rh nanotip to $H_2$ and $O_2$ gas mixtures in the $10^{-7}$ to $10^{-5}$ mbar range, adjustable by dosing via precision leak valves. The composition of the gas phase was monitored by a mass spectrometer, with the reaction product water continuously being pumped off by a turbomolecular pump.

The ongoing catalytic $H_2$ oxidation reaction was visualised in situ by FEM, and the reaction kinetics was analysed based on the recorded video-images (kinetics by imaging approach[38]): both the catalytic activity and work function (and thus the FEM image brightness) depend on the surface concentration of reactants. The Langmuir–Hinshelwood mechanism[43,44] of the reaction and significant differences in the local work function and thus in the local image brightness allow the distinction between the catalytically active surface regions, with a low coverage of both hydrogen and oxygen (lower work function, bright image contrast), from the catalytically inactive, oxygen covered regions (higher work function, dark image contrast). By isothermal cyclewise variation of $p_{H2}$ at constant $p_{O2}$, kinetic transitions between the catalytically inactive and active steady states ($\tau_A$) and vice versa ($\tau_B$) can be triggered. The resulting rapid changes of the surface coverage, which accompany these transitions, are reflected in the FEM image intensity and can be used for determination of the $\tau_A$ and $\tau_B$ transition points via variation of pressure[32,37] or temperature[45]. The evaluation procedure is described in detail in Supplementary Note 2. Figure 2a, b shows the course of the FEM image intensity for the Rh and Rh/La systems during cyclewise variation of $p_{H2}$ at constant $T = 473$ K and $p_{O2} = 7.7 \times 10^{-7}$ mbar.

At a low ratio of $p_{H2}$ to $p_{O2}$, the inactive steady state (oxygen covered surface) characterised by high work function and low FEM intensity is established. When the increasing hydrogen partial pressure reaches a certain threshold ($\tau_A$), the adsorption of hydrogen becomes favoured, and the system switches to the catalytically active steady state (low surface coverage, high FEM intensity). Such $\tau_A$ transitions are accompanied by fast reaction-diffusion fronts spreading over the entire field of view and are reflected as steep jumps in the local FEM intensity curves (Fig. 2a, b). Consequently, upon decreasing the hydrogen partial pressure, the reverse kinetic transition ($\tau_B$) to the inactive state occurs. This transition, in contrary, typically appears as a continuous successive intensity decay without a global reaction-diffusion front[37,44]. This effect, well visible in Fig. 2a, is intensified in the La/Rh system (Fig. 2b). Apparently, La promotes the competitive replacement of the thinned out $H_{ads}/O_{ads}$ layer by a dense oxygen layer. Importantly, the respective $\tau_A$ and $\tau_B$ transitions occur at different $p_{H2}$ for the Rh and La-promoted Rh surface (cf. Fig. 2a, b). The hysteresis-like behaviour of the FEM intensity curves manifests the bistability of the $H_2$ oxidation on Rh[37,44,46,47] which is maintained also in the present case of La-modification.

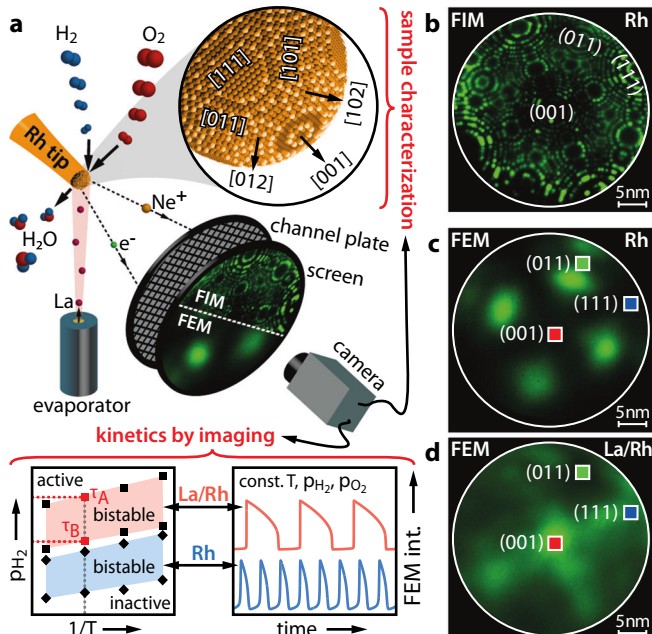

**Fig. 1 | Experimental approach. a** Schematic of the experimental setup and sample geometry: In FIM and FEM, field emitted ions and electrons, respectively, form a point projection image of the sample surface. Using the adsorbate dependent FEM image intensity, in situ kinetic studies can be performed. A La-evaporator produces sub-monolayer coverages of La on the Rh surface. **b** FIM image of the [001]-oriented Rh nanocrystal, obtained at T = 77 K using Ne$^+$ ions. Low-Miller-index facets are indicated. **c** FEM image of the same field of view as in **b** with square ROIs placed on the indicated facets. **d** FEM image of the same Rh nanocrystal and field of view as in **b** and **c**, but with an additional 0.2 ML coverage of La.

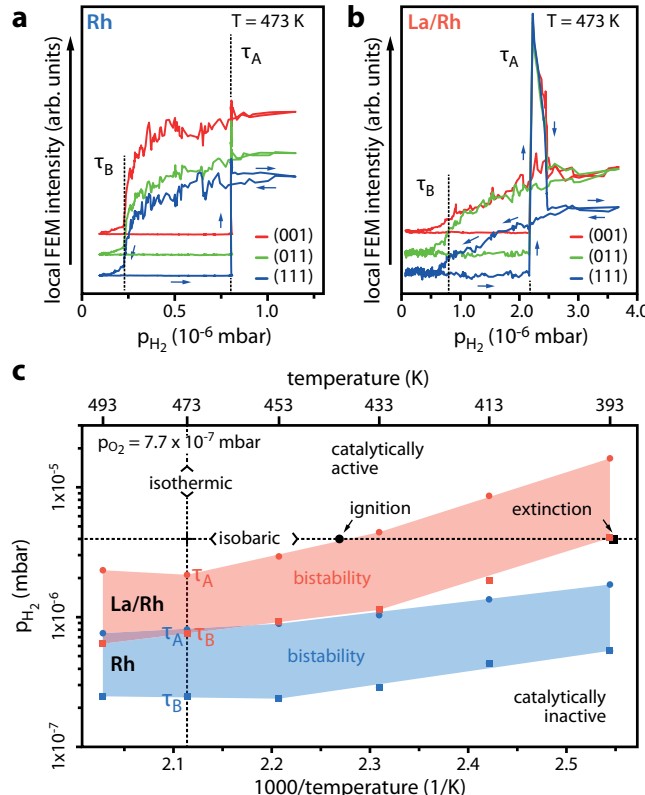

**Fig. 2 | Catalytic $H_2$ oxidation on the La-free and La (0.2 ML) covered Rh nanotip at constant $pO_2$ of $7.7 \times 10^{-7}$ mbar. a** Isothermal hysteresis curves measured locally for the ROIs placed on Rh(001), Rh(011) and Rh(111) nanofacets during cyclewise variation of $p_{H2}$ at T = 473 K for Rh. **b** The same as in **a**, but for La sub-monolayer covered Rh. **c** Kinetic phase diagrams for $H_2$ oxidation in the temperature range from 393 K to 493 K constructed for Rh (shaded blue) and La/Rh (shaded red) from isothermal hysteresis measurements. The isobaric ($p_{H2}$ of $4.0 \times 10^{-6}$ mbar) ignition and extinction temperatures for cyclewise variation of the temperature are marked, matching and corroborating the isothermal phase diagram.

By performing hysteresis experiments for different constant temperatures, but the same constant $p_{O2}$, the kinetic transition points $\tau_A$ and $\tau_B$ determined from each recorded FEM intensity curve can be plotted in the $p_{H2}/T$ parameter space as a kinetic phase diagram. The use of the term "phase diagram" for such plots is justified by the analogy to equilibrium thermodynamics[48,49]:

The crucial role in both, equilibrium and nonequilibrium phase transitions, is played by cooperative phenomena forming, e.g., ordered structures in an equilibrium, and self-organising dissipative structures in a nonequilibrium situation. From this point of view, the appearance of instabilities, bifurcations or kinetic oscillations is related to the appearance of different competitive reactive"phases"[50].

Figure 2c shows kinetic phase diagrams for constant $p_{O2} = 7.7 \times 10^{-7}$ mbar in the temperature range from 393 to 493 K for both La-free and La-modified systems. Apart from isothermal experiments, it is also possible to induce kinetic transitions by isobaric temperature variations (ignition/extinction experiments). The results of such an isobaric measurement are exemplarily plotted in Fig. 2c. As expected, the resulting transitions coincide with those from the isothermal experiments indicating that the ignition and extinction temperatures of the catalytic reaction can be read out from the isothermally obtained kinetic phase diagram, a procedure established e.g. for the CO oxidation reaction[38,51,52].

The shape of the bistability region in Fig. 2c reveals the temperature dependent kinetics of the hydrogen oxidation. At low

temperatures, kinetic transitions occur at higher $p_{H2}$ indicating oxygen being favoured in the competitive adsorption. While this trend is also present in the La/Rh system, a significant shift to higher $p_{H2}$ is observed. This effect can be traced back to the higher binding energy of oxygen on a Rh surface promoted by a sub-monolayer of La. It is known that electropositive adsorbates such as alkali or alkaline earth metals favour the adsorption and dissociation of molecular oxygen[1,3,53–55] and increase the binding energy of adsorbed oxygen[55,56]. Such effect can thus be expected for the electropositive adsorbate lanthanum, which reduces the surface work function to a degree quite comparable with alkali or alkaline earth adsorbates[21,39,40,57].

## Self-sustaining oscillations

In addition to the steady state modes, $H_2$ oxidation on Rh can also operate in a self-sustained oscillatory mode at particular constant external parameters ($p_{H2}$, $p_{O2}$ and T), where the system periodically switches between the inactive and active state[46,58–60]. The feedback mechanism governing the oscillation cycle is based on the formation/depletion of subsurface oxygen which modulates the sticking coefficient of oxygen[31,60,61]. The oscillation frequency is very sensitive to local surface crystallography, as the activation energy for the formation/depletion of subsurface oxygen is strongly dependent on the local atomic surface roughness and on the presence of step and kink sites[61,62]. This makes the oscillating $H_2$ oxidation a sensitive tool for studying surface heterogeneity[60], even down to the nanoscale[32]. In addition, such reaction mode provides a possibility to study a close sequence of repeating kinetic transitions which occur at the same experimental conditions at the atomically same surface configuration, as proven by microscopic monitoring.

In this study, we used this possibility to reveal the origin of local pacemakers in $H_2$ oxidation, which are particular local surface atomic configurations responsible for initiating kinetic transitions. This also shed light on the role of a La-promotor in kinetic instabilities in this reaction.

On the current Rh nanotip, both without and with La, self-sustained oscillations were observed at constant external parameters of $p_{H2} = 5.0 \times 10^{-6}$ mbar, $p_{O2} = 4.4 \times 10^{-6}$ mbar and T = 453 K. Figure 3a exemplarily shows FEM video frames acquired during different stages of the oscillatory cycle: the inactive state (frame 1), the transition to the active state (frame 2) displaying primary oscillation peak P1, and the instable proceeding catalytic reaction (frame 3). Such primary peaks, labelled P1 to P4 in Fig. 3b showing the oscillating FEM intensity curves, occur with a repetition frequency of ~ 0.5 mHz.

Following each primary peak, fast low-amplitude oscillations (secondary peaks $S_n$) lasting approximately half of the primary period occur before the image intensity returns to its initial stable level, corresponding to the inactive state. These follow-up secondary peaks arrange into particular fast-paced oscillating intervals. The bottom panel of Fig. 3a exemplarily shows a magnified section of such an interval consisting of secondary peaks, e.g., $S_{P4}$, which repeat with a frequency of ~17.5 mHz. Despite the presence of multiple facets of differing crystallography, the observed oscillations are synchronised over the whole tip apex as usually observed for oscillating reactions in a nm-sized system[28,63–65]. The synchronisation results from diffusional coupling via adsorbed highly mobile hydrogen unless its breakdown under specific reaction conditions allows for different frequencies of individual nanofacets[31,34].

The experiment was repeated for the La-precovered surface under the same external conditions. Figure 3c shows FEM video frames of the respective inactive state (frame 1), the transition to the active state (frame 2) with a primary peak L2 and the instable proceeding reaction (frame 3). Figure 3d shows the corresponding FEM intensity curves oscillating with a frequency of ~0.5 mHz, again with the oscillations being synchronised over the whole tip apex. In contrary to the La-free Rh surface, no pronounced secondary peaks were observed.

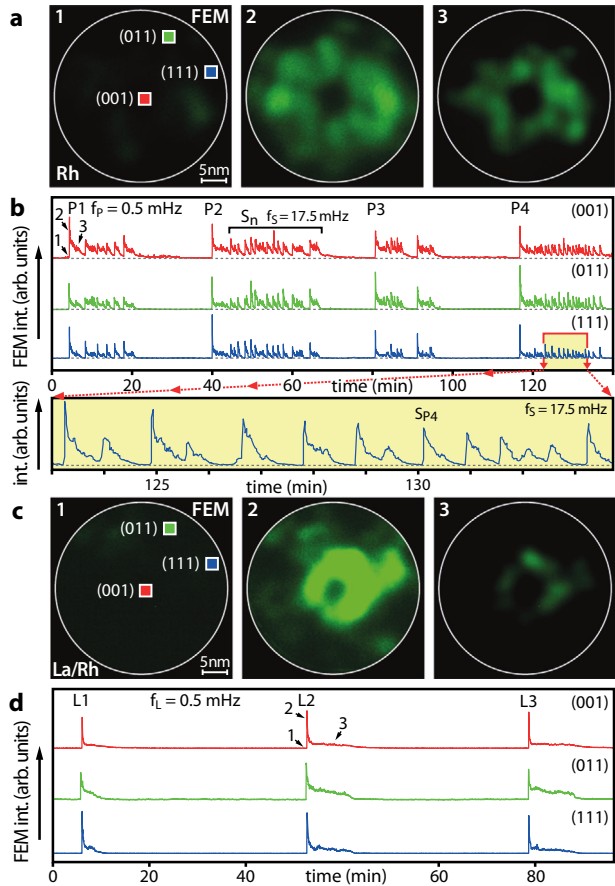

**Fig. 3 | La-coadsorption effect on the oscillating catalytic H₂ oxidation at constant $p_{H2} = 5.0 \times 10^{-6}$ mbar, $p_{O2} = 4.4 \times 10^{-6}$ mbar and T = 453 K. a** In situ recorded FEM video frames of the La-free Rh surface. **b** Local FEM intensity curves registered within ROIs ($2 \times 2$ nm²) placed in (001), (011) and (111) regions. Four consecutive primary peaks are labelled P1–P4 and secondary $S_n$ peaks are exemplarily indicated. The time points corresponding to video frames 1–3 in **a** are indicated. The magnification of the yellow marked time interval containing secondary peaks ($S_{P4}$), is shown in the bottom section of **b**. **c** The same as **a** but for the La-promoted (0.2 ML) Rh surface. **d** The same as in **b** but for the La-promoted surface. The time points 1–3 correspond to video frames in **c**.

## Two types of pacemakers identified

On the La-free Rh surface, the registered timeseries indicate the presence of two superimposed oscillations, differing in their frequencies by a factor of about 40. This suggests the existence of two types of pacemakers, i.e., particular local surface atomic configurations which initiate the local kinetic transitions and the nucleation of reaction fronts at their own pace[32]. In contrast, on the La-promoted surface the observed timeseries suggest only one acting type of pacemaker. The coincidence of the repeating frequency of the P peaks (La-free surface) and the L-peaks (La-promoted surface) suggests at first glance that the pacemakers responsible for the primary oscillations remain largely unaffected by La, while the other pacemaker type, which causes the fast secondary oscillations (S peaks), is strongly impeded by La coadsorption.

To fundamentally understand this behaviour and to identify the pacemakers that initiate the P- and S-peaks, it is required to investigate the local spatio-temporal behaviour in detail. The analysis of the reaction-diffusion front propagation can be used to locate where a front nucleates, i.e., to pin down the local pacemakers. The path of the reaction front, transporting the local kinetic transition from the inactive to the active state across the sample surface, can be traced by the recently developed method of "transition point tracking" (TPT)[32]. As a

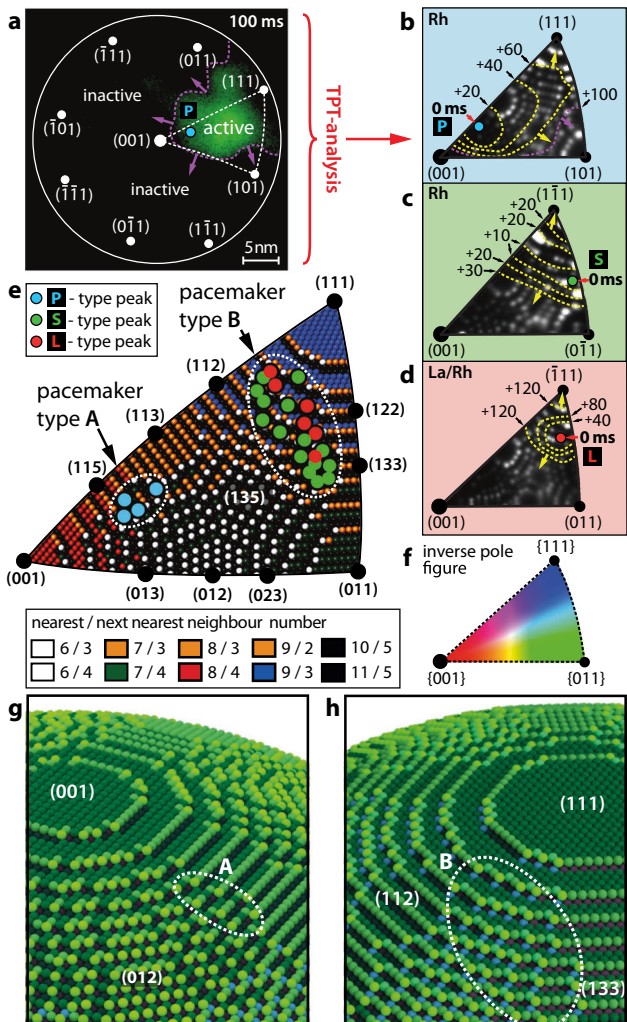

**Fig. 4 | Analysing oscillatory transitions from the inactive to the active state. a** Difference image illustrating the propagation of the active region (peak P2 in Fig. 3b): Active/inactive regions, the reaction front position 100 ms after the nucleation (purple line) and local crystallographic orientations are indicated. **b–d** Consecutive exemplary local reaction front positions during transitions of type P, S and L. The corresponding underlying FIM image segments illustrate the local crystallography. **b** First 100 ms of the P2 transition in the dotted triangle indicated in **a**. **c** First 30 ms of the $S_{P4}$ transition in Fig. 3b. **d** First 120 ms of the L2 transition in Fig. 3d. **e** Different pacemaker types positioned as coloured discs on a corresponding ball model segment matching the inverse pole figure. To illustrate the local atomic corrugation, the individual atoms are colour-coded according to their nearest and next-nearest neighbour numbers. **f** Inverse pole figure containing all crystallographic directions, representing the segments shown in **b–e**. **g** Ball model illustration of the pacemaker A region. **h** Same as in **g**, but for the pacemaker B.

reaction front is basically a travelling continuum of kinetic transition points, the propagation of the reaction front reflects the spatial evolution of the kinetic transition (Fig. 4a).

TPT analyses the progression of the intensity of each of the 86,644 pixels of the FEM video frames recorded during the front propagation. It thus allows determining the local transition time point for each surface location with high spatial resolution. In this way, the TPT analysis reveals the positions of pacemakers by localisation of the front nucleation centres with nm-precision[32] (see Supplementary Note 2 and Supplementary Fig. 2 for details).

The oscillatory kinetic transitions on the La-free surface (Fig. 3b) were evaluated by TPT and can be characterised as follows: primary P peaks (Fig. 3b, top panel) nucleate at pacemakers located on the

perimeter of the {113} facets, being positioned on the [001]-oriented terraces of these stepped regions. The pace-making behaviour is exemplarily shown in Fig. 4b for the primary peak P2. The same data reveal a second transition type, which forms the fast local secondary peaks of type $S_n$. The evaluation of such an S-transition is shown in Fig. 4c for the peak $S_{P4}$ (Fig. 3b, bottom panel) and was also performed for multiple other S-type peaks. The analysis localises the pacemakers responsible for S-type peaks at {111} vicinals exhibiting terraces of [111] orientation (Fig. 4e, f).

Summing up the TPT analysis of the repeating oscillating transitions on the La-free Rh surface, we note the existence of two different types of pacemakers which differ substantially in their properties: P-type, initiating the primary transitions, and S-type, initiating the fast secondary transitions and showing up as fast oscillations with small amplitude. The differing locations of the P- and S-type transitions are summarised in the Fig. 4e, demonstrating that locations of both types of pacemakers are subject to a certain "scattering". This can be understood by considering that the facets of the Rh nanotip are only a few nm in size. The adsorption, diffusion and reaction are stochastically influenced processes, with increasing contribution of the stochastic component when the system size decreases, eventually leading to an increasing role of fluctuations in kinetic transitions[66]. This may lead to the slightly varying positions of particular pacemakers, which nevertheless retain their local crystallographic properties that were described above.

For the La-promoted Rh surface, under the same experimental conditions, only one type of pacemaker was identified (Fig. 4d). The pacemakers which initiate the oscillatory transitions on the La-promoted surface (L-type) were also found by TPT on the {111} vicinal regions, i.e., at locations where S-type pacemakers were detected on La-free Rh (Fig. 4e). This means that addition of La to the reaction system leads to a surprising phenomenon: the S-type pacemakers, which were responsible for the fast S-type transitions on the La-free surface, take over the role of the primary pacemakers in La-modified oscillations. Consequently, the P-type pacemakers responsible for the primary oscillations on the La-free surface are disabled and fast secondary oscillations do not occur on the La-promoted surface.

Thus, two types of pacemakers labelled as A and B (Fig. 4e, g, h) can explain the entire range of oscillating phenomena in $H_2$ oxidation on La-free and La-promoted surfaces of a [100]-oriented Rh nanotip, i.e., they are responsible for all three types of oscillatory kinetic transitions observed under the present conditions. This unexpected behaviour calls for an atomistic explanation, however.

## Micro-kinetic modelling

To rationalise the present experimental findings and to reveal the role of coadsorbed La in the reaction, micro-kinetic model simulations were performed. We utilised our recently developed model system of coupled oscillators[33,34], based on the single-oscillator model introduced by McEwen et al. for field-induced oscillations in $H_2$ oxidation[58,59] as observed in FIM applying an electrostatic field >10 V/nm for imaging[58,59]. In the present FEM experiments, the applied electric field is much lower (<5 V/nm) and of opposite direction, which excludes a field-induced redistribution of electron density near the surface, ruling out a possible field effect on the adsorption of reactants. The absence of field effects on $H_2$ oxidation on Rh in an FEM was recently directly proven experimentally using a pulsed field with varying duty pulses[34] (see Supplementary Note 3 for details). For this reason, the field related terms in the model equations were omitted.

Based on the experimental observations and knowledge of the crystallographic properties of the present Rh nanotip, we translated the involved surface regions into a network of 25 oscillators. Five different oscillator types were used to represent the different crystallographic surface regions and were arranged as a symmetric grid mimicking the tip apex symmetry (Fig. 5a). Different oscillator types represent sets of the low-Miller index facets {100}, {110} and {111} present on the real nanotip surface (Fig. 4a) and two highly stepped regions A and B representing the pacemakers, differing by their atomic configuration as shown in Fig. 4e–h. The surface structure of particular domains, related to the local atomic surface roughness, is reflected by the sticking coefficient of oxygen and the activation energy of the formation of subsurface oxygen $E_{ox}$. The latter governs the periodic formation and depletion of subsurface oxygen, which serves as the feedback mechanism of the oscillations[31,60,61,66].

The $E_{ox}$ distribution across the different types of oscillators is shown in Fig. 5a. Surface diffusion of hydrogen, which provides communication between adjacent oscillators, was incorporated into the model equations via a diffusion term. The surface diffusion of oxygen could be neglected, because under the present conditions it is by three orders of magnitude slower than that of hydrogen[33,34,67].

To better understand the role of multiple pacemakers, we modelled the oscillation behaviour of the system under the same conditions, but with artificially disabled coupling. Figure 5c shows the resulting behaviour of the same oscillators as in Fig. 5b but with coupling "switched-off". Pacemaker A exhibits low frequency oscillations while B keeps its frequency close to that in the coupled system. In contrast, the low-Miller-index oscillators remain inactive. Comparing

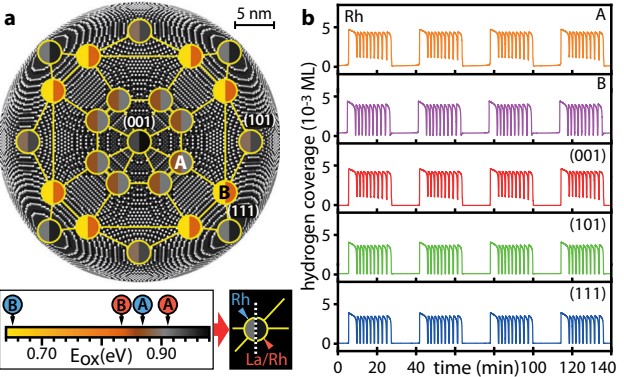

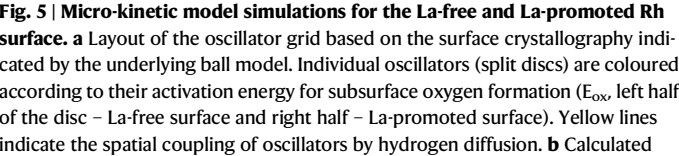

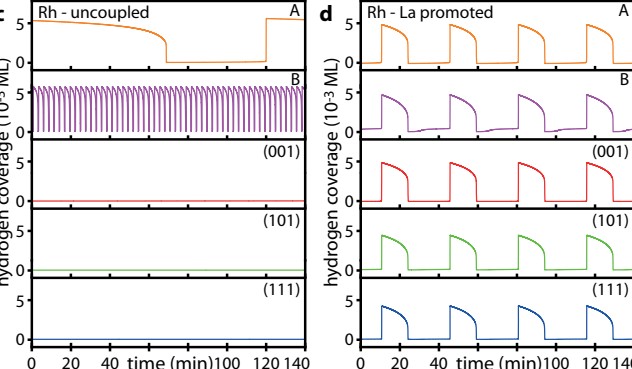

**Fig. 5 | Micro-kinetic model simulations for the La-free and La-promoted Rh surface. a** Layout of the oscillator grid based on the surface crystallography indicated by the underlying ball model. Individual oscillators (split discs) are coloured according to their activation energy for subsurface oxygen formation ($E_{ox}$, left half of the disc – La-free surface and right half – La-promoted surface). Yellow lines indicate the spatial coupling of oscillators by hydrogen diffusion. **b** Calculated timeseries of the hydrogen coverage for the coupled oscillators A, B, (001), (101) and (111) at constant $pH_2 = 5.0 \times 10^{-6}$ mbar, $pO_2 = 4.4 \times 10^{-6}$ mbar and $T = 453$ K for the La-free surface. **c** Behaviour of the oscillators from **b** without coupling revealing their natural frequencies. **d** Calculated timeseries for the La/Rh system under the same conditions as **b**.

the coupled and uncoupled systems (cf. Figure 5b, c) shows that the low-Miller-index oscillators play solely the role of excitable regions which are entrained via coupling with pacemakers A and B in the coupled system.

The pacemaker region A generates oscillations via formation of subsurface oxygen which reduces sticking of oxygen, thus switching the local competitive coadsorption to the (catalytically active) hydrogen side. The reaction-caused depletion of the subsurface oxygen depot switches the system back to the initial state closing the oscillation cycle (details of the oscillation mechanism are presented in Supplementary Note 4). The pacemaking role results from the fact that under the present conditions, the external parameters ($p_{H2}$, $p_{O2}$, T) put the region A is in its oscillating state, contrary to its surrounding.

During the oscillation cycle, pacemaker A temporarily becomes a source of adsorbed hydrogen upon its transition to the active state. This mobile hydrogen diffuses into surrounding regions in addition to the hydrogen supplied from the gas phase. This temporarily shifts the "positions" of these regions in the parameter space similarly as increasing the hydrogen pressure would do. The region B then finds itself in its natural oscillatory state and takes over the leading role due to its much faster oscillation frequency. The rest of the surface becomes more excitable and thus affine to the entrainment of fast oscillations generated by B. During the relatively long "active state" half-period of A, pacemaker B triggers transitions on the remaining oscillators by providing a modulated hydrogen diffusion supply.

In this way, the complex temporal pattern observed in the experiment can be rationalised as a result of a pacemaker induced communication via hydrogen diffusion, which generates a shift in the parameter space due to the extra hydrogen provided via diffusion from the catalytically active regions.

The coadsorption of electropositive adsorbates is known to increase the binding energy of oxygen[53,55,56], an effect which was also corroborated for La adsorption by initial exploratory density functional theory (DFT) calculations (see Supplementary Note 5 for details). Therefore, the presence of coadsorbed La was considered in the model by modifying the corresponding kinetic parameters for oxygen adsorption. Additionally, the lateral oxygen-lanthanum interaction is expected to increase the activation energy for subsurface oxygen formation, particularly when La partially occupies the low-coordination step and kink sites favourable for subsurface oxygen formation[62]. In the model simulations for the La-promoted Rh surface, the local $E_{ox}$ values were thus higher than those for the La-free Rh (for details see Supplementary Note 4 and Supplementary Fig. 4). Results of the calculations for the La-promoted Rh surface are shown in Fig. 5d: the modified system still exhibits globally synchronised oscillations in the form of periodic oscillations but this time with a singular pacemaker.

## Discussion

To explain the behaviour of a La-modified surface we refer to the known properties of rare earth adsorbates on transition metal surfaces. Any metal surface exhibits its particular spatial corrugation of the electron density which is defined by the surface atomic structure. Adsorption of electropositive adsorbates shifts the electron density distribution further away from the surface by an extent depending on its structure[68–70]. The evident consequence of this global effect is a lowering of the work function, the less evident one is the impact on the adsorption behaviour of reactants, such as oxygen, in a catalytic reaction. The coadsorption-induced increase in the binding energy of oxygen and thus in the sticking coefficient are important components in the promoting effect of electropositive coadsorbates, such as alkali[71–73], alkaline earth[74] and rare earth[20] elements.

In the present case of $H_2$ oxidation on Rh, such redistribution of the electron density, caused by La-coadsorption and usually treated globally, shows a rather unexpected local impact, drastically influencing the behaviour of the local reaction pacemakers and thus the course of kinetic transitions.

This can be understood when one considers that the La-coadsorption caused modifications of the adsorption kinetics directly affect the reaction kinetics: an increase in the sticking coefficient of oxygen corresponds to a virtual pressure increase. The surface regions working as type A pacemakers are then shifted out of the oscillating regime, i.e., pacemakers A are disabled. Since the degree of modification depends on the particular atomic configuration, the pacemaker role of regions B apparently remains less affected. It is known from earlier studies, that an increase in oxygen pressure slows down oscillations in $H_2$ oxidation on Rh[66]. This explains the significant slowdown of oscillations generated by pacemaker B on the La-promoted surface. Summarising, the communication of two types of reaction pacemakers in the hydrogen oxidation reaction leads to a complex oscillating temporal pattern which was not observed before: fast oscillations generated by one type of nano-pacemakers are modulated by slow oscillations of another type. The key to such an unexpected effect lies in the nm-size of the studied system: small amplitude variations in surface diffusion are usually smeared out in accord with Fick´s law[75] and can therefore hardly influence the kinetic behaviour of neighbouring regions in macro- or mesoscale studies. In the present case of nm-sized facets separated just by atomically wide edges, variations of hydrogen diffusion cannot be neglected and contribute significantly to the spatio-temporal behaviour of the reaction nanosystem.

In the present study, the observations above were analysed for self-sustaining oscillating transitions, but, of course, they are also valid for kinetic transitions induced by variations of partial pressure or temperature (catalytic ignition): findings made for the oscillating mode of the reaction also agree with steady state observations of the shift in the kinetic phase diagram.

As the Langmuir−Hinshelwood mechanism of the catalytic $H_2$ oxidation has been validated for a wide pressure range up to ambient conditions[76,77], the observed La-modification effect should also be relevant for technological applications where kinetic transitions are involved, as e.g., response of hydrogen sensors[78], onsets of elimination of hydrogen traces in the oxygen stream of electrolysers[79], recombination of hydrogen off-gas from a fuel cell[80] and elimination of hydrogen formed in an accident inside water-cooled nuclear facilities[81]. Furthermore, the observation that electropositive promotors essentially influence the local reaction pacemakers may open new perspectives for research in heterogeneous catalysis and hydrogen-based chemical sensors.

## Methods

### Experimental setup

All experiments were conducted in an FIM/FEM setup, equipped with a sample holder which allows operation in the temperature range from 77 to 900 K, a channel-plate/screen assembly for imaging with gas ions, e.g., Ne⁺ (in the FIM mode) or by electrons (in the FEM mode), a high purity gas supply hub (O₂: 99.999% Messer, H₂: 99.97% Linde), a quadrupole mass spectrometer (MKS e-Vision 2), and a miniature home-made Knudsen-type effusion cell as a La-evaporator.

### Sample preparation

The Rh nanotip was fabricated by electrochemical etching of a Rh wire (0.1 mm, Mateck, 99.99%), in a sodium nitrate/sodium chloride (4:1) salt smelter followed by shaping and cleaning the apex surface by field evaporation under UHV conditions at 77 K. The shaping by field evaporation was controlled in situ by FIM imaging with atomic resolution. For experiments on the La/Rh system, La was deposited by shutter-controlled evaporation and the La-coverage was controlled by the directly measured work function of the La-covered Rh surface. Details on the characterisation of the La coverage are given in Supplementary Note 1.

## FIM/FEM measurements

The UHV chamber of the FIM/FEM setup was operated as a flow-reactor for catalytic hydrogen oxidation reaction on unpromoted and La-promoted Rh. The constant gas phase composition in the $10^{-6}$ mbar range during the FEM monitoring of the reaction was controlled by precision leak valves and verified by a quadrupole mass spectrometer. Particular attention was paid to a high precision control of the Rh-nanotip temperature, using a newly developed automatised sample temperature control system[45]. Thereby, a thermocouple (Ni/CrNi) directly spot-welded to the shank of the Rh nanotip served as a temperature sensor. FIM images during the sample preparation and the FEM video frames during the ongoing hydrogen oxidation were recorded by a CCD camera (Hamamatsu C13440).

## Data processing

The recorded FEM video files were analysed by an automated algorithm that processes and evaluates the local FEM intensity timeseries. Oscillating and non-oscillating timeseries were distinguished via the analysis of the autocorrelation-function and the oscillation frequencies were determined by Fourier analysis. Spatial correlations between local FEM intensities were determined by computing the respective cross-correlation functions.

## Transition Point Tracking (TPT)

The reaction pacemakers serve as the origin points of the front nucleation, and the propagating fronts can be interpreted as a continuum of consecutive local kinetic transitions. Transition point tracking determines the time point of the local transition from the inactive to the active state for each pixel via automatic analysis of the local timeseries. This allows a precise reconstruction of the reaction front propagation despite the limited time resolution. Details on Transition Point Tracking are given in Supplementary Note 2.

## Micro-kinetic model simulations

Micro-kinetic model simulations were carried out based on the Langmuir–Hinshelwood mechanism of the hydrogen oxidation reaction on rhodium and the formation/depletion of subsurface oxygen as the feedback mechanism governing the self-sustained oscillations. Particular nanofacets of the Rh-nanotip were modelled by a grid of 25 oscillators. Spatial coupling via hydrogen diffusion was considered by a corresponding term in the model simulations. Details regarding the micro-kinetic simulations are presented in Supplementary Note 4.

## DFT calculations

DFT-PBE calculations were carried out using slab models with periodic boundary conditions. The infinite Rh(111) slab consisted of 48 atoms per unit cell, with the top layer relaxed. Oxygen adsorption was probed in the presence or absence of an added lanthanum adatom, sitting at a threefold hollow site. More details on methodology and the model are provided in Supplementary Note 5.

## Data availability

The FEM video data have been deposited in a Zenodo repository under https://doi.org/10.5281/zenodo.8155170. The parameters used for modelling are provided in the Supplementary Information.

## Code availability

All the code or mathematical algorithm files within this paper are available from the corresponding authors upon reasonable request.

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

## Acknowledgements

This work was supported by the Austrian Science Fund (FWF P 32772-N and SFB TACO F81-P08 (G.R.)).

## Author contributions

M.R., J.Z. and C.W. performed the FEM/FIM experiments, M.R. and J.Z. carried out the processing and analysis of the FEM/FIM video-data, M.R. performed the micro-kinetic modelling, A.G. carried out the DFT calculations, Y.S. supervised the experiments, Y.S. and G.R. advised in the analysis of the experimental data. M.R., J.Z., Y.S. and G.R. prepared the manuscript. All authors contributed to the discussion and approved the manuscript and supporting information.

## Competing interests

The authors declare no competing interests.
