## [Peer Review File · Nature Communications]

REVIEWER COMMENTS

Reviewer #1 (Remarks to the Author):

This manuscript presents the field electron microscopy (FEM) for in situ monitoring the La-mediated local effect of Rh catalysts for H₂-O₂ oxidation reaction. The methods and corresponding results are impressive. However, the structure effect of La to mediate the H₂ or O₂ adsorption is insufficient. The kinetically relevant steps and dominated intermediates such as hydroxy species were not well investigated. The quality of data presentation/interpretation needs improvements and validation.

1. The authors clarified the various effect of alkali promoters in the Introduction section, have the authors considered the effect of alkali on the acid-base properties of support.
2. The authors build on in situ characterization. However, it should be noted that none of the characterization was conducted operando i.e while the reaction runs on the catalyst and gaseous products are thoroughly analyzed while the catalyst is being characterized. Can the authors provide the product distributions in addition to the FEM intensity ?
3. Can the authors provide the evidence for the relationship between the measured work function and La coverage? Is there formation of Rh-La bimetallic assemblies? Would the assemble effect or strain effect between Rh and La influence the work function?
4. Is the reaction sensitive to O₂ pressure and coverage, i.e., would the reaction mechanism change if the H₂-O₂ reaction was performed at the high pressure ($p_{O_2} = 7.7 \times 10^{-7}$ mbar) and coverage of O₂ or the very low coverage of O₂?
5. Would the H₂ be adsorbed on the surface of La? Previous report reveled the stabilization effect of H species by alkali ions (Nature Catalysis volume 3, pages703–709 (2020)).
6. How did the authors determine the position of τ_B for the La/Rh systems?
7. There are many typos and edit errors for the references. Please carefully check them out.

Reviewer #2 (Remarks to the Author):

Recommendation: Minor Revision

This work investigated the effects of co-adsorbed La in Rh nanotip on the H₂ oxidation with in-situ monitoring of FEM. In-situ characterization of catalytic reaction and structural evolution is of great interests and poses significant difficulties in both experimental and theoretical approaches. The authors

identified different pacemakers between La-mediated Rh surface and La-free Rh surfaces and revealed the role of La using micro-kinetic modeling simulations. The reviewer thinks the work is a good extension of their previous works such as 10.1021/acscatal.3c00060, 10.1038/s41467-021-26855-y, 10.1126/science.abf8107, on in-situ characterization of FEM for catalytic reactions and provides some new insights on the role of promoters in catalysis. Below are some minor comments for the further improvement of this work.

1. The authors investigated the La-induced phase changes. How could this finding differ from as-synthesized catalysts of La-Rh nanoparticles?
2. The authors utilized microkinetic modeling to understand the experimental observations of different pacemakers. They adjusted the kinetic parameters of oxygen adsorption to incorporate the effects of electropositive La. What would be the effects of species with different electronegativity? Atomistic simulation with first-principles calculations may elucidate the effects of binding affinity change and predict local surface atomic configurations.

Reviewer #3 (Remarks to the Author):

The manuscript reports the tuning of the bistable reaction states with a La-mediated local catalytic effect. The experimental measurements are based on in-situ FEM imaging, from which the variation of the image contrast is used to extract the reaction kinetics and catalytic states of the catalyst surface. The authors provide convincing experimental results to show the effect of the La deposition on changing the bistability pattern of the reaction, and the fundamental insights are of interest to the catalysis community. The authors are suggested to address the following main issues:

1. introduction: some information about why the Rh/La system is of particular interest and importance would be necessary.
2. sample preparation: how the atomically clean Rh tip is ensured, particularly on the different facets? Is there any direct evidence such as by XPS or AES to confirm the surface cleanliness? How are the three facets ((001), (011) and (111)) of the Rh tip ensured to have the same coverage of La from the evaporation deposition?
3. Fig. 2 shows the measurements in determining the p_{H_2} and T range between the active and inactive states. It would be helpful to show how many cycles of the cyclewise variations of p_{H_2} and T are performed to show the repeatability. Are there any differences in the hysteresis curves between different cycles, and does the catalyst (and promoter La) show any kind of deactivation that may induce changes in the hysteresis curves? The experiments are performed at elevated temperatures that may

result in significant surface diffusion of La adatoms, for which the surface coverage of La may change over time. Any results and/or discussion of this dynamic nature of the catalyst surface facets on the evolution of the hysteresis curves would be interesting.

4. Fig. 2 and 3 show that all the three facets ((001), (011) and (111)) have the synchronized oscillations between the active and inactive states (in terms of p_{H_2} and T), is this an artifact from the experiment measurements (imaging intensity) due to the close proximity of the three facets or an intrinsic behavior? If it is the latter, what is the specific reason for inducing the synchronized oscillations for the three facets with different surface structures and even different La coverage? How do the bi-stable reaction states (or self-organizing dissipative structures) depend on the local crystallography and La coverage of different facets?

5. Fig. 2c shows the shift of the bistability region to the higher p_{H_2} in the La/Rh system. This shift is attributed to the higher oxygen energy due to the La promoter effect. If this is true, the bistability region may also depend on the surface coverage of La, any experimental measurements of the bistability region for different La coverages would make a more convincing case.

6. For the measurements shown in Fig. 3, what is the surface coverage of La for the (001), (110) and (111) facets? Do the primary peaks depend on La coverage? And why do the three facets show the same oscillation frequency? This seems to contradict with the statement “the oscillation frequency is very sensitive to local surface crystallography” on page 6.

7. It is proposed in Fig. 4 and in the discussion section that the S-type pacemakers can be modified by the La coverage. Any results from experiment results with different La coverages would strengthen this point. It is proposed in Fig. 4e that the two types of the pacemakers are related to the high-index facets that may be unstable and evolve over the time at the elevated temperature. Are La also deposited onto these high index facets and how stable of the La adatoms on these high-index facets at the elevated temperature?

8. The measurements are performed at the extremely low pressure regime ($\sim 10^{-6}$ mbar), are the results relevant to the realistic catalytic reactions at a much high pressure? Is there a pressure gap regarding the effect of the La promoter on tuning the bi-stable reaction states?

Reviewer #1

This manuscript presents the field electron microscopy (FEM) for in situ monitoring the La-mediated local effect of Rh catalysts for H₂-O₂ oxidation reaction. The methods and corresponding results are impressive.

We thank the Reviewer for this encouraging comment.

However, the structure effect of La to mediate the H₂ or O₂ adsorption is insufficient. The kinetically relevant steps and dominated intermediates such as hydroxy species were not well investigated. The quality of data presentation/interpretation needs improvements and validation.

At submonolayer coverages as studied in the present work, La, like other electropositive coadsorbates, forms open wide-meshed structures on transition metal surfaces (Refs. 21, 40, 41, 42 in the revised manuscript). At such coverages, the effect on H₂ or O₂ adsorption is mainly of electronic nature, due to the shift of electron density away from the surface (towards the gas phase). The distribution of the electron density on the outside of the surface is homogenized by the electron-electron interaction, therefore the resulting effects are well described by the jellium model (see e.g. Lang, N. D. *Solid State Commun.* **9** (1971) 1015; Nørskov, J. K. *Surf. Sci.* **137** (1984) 65; Lang, N. D., *Surf. Sci.* **150** (1985) 24; Nørskov, J. K. et al. *J. Vac. Sci. Technol. A* **3** (1985) 1668; Nørskov, J. K. in: *The Chemical Physics of Solid Surfaces* **6**, (1993) 1-27). A La-structure effect is therefore negligible, and peculiarities related to the Rh surface structure are not expected due to the equalized La-coverage across the nanotip surface (see points Rev. #1.3 and Rev. #3.6 below). An insight into overlayer structures on the nanofacets of a nanotip (or nanoparticles) during an ongoing reaction is far beyond the possibilities of modern surface analysis techniques. All the kinetically relevant steps and intermediates (solely hydroxyl, see e.g. Ref. 45) in the H₂ oxidation on Rh are well known and remain the same in the presence of a submonolayer of an electropositive adsorbate. Based on the suggestions of the Reviewer and the comments of the other two Reviewers (see below), an additional chapter regarding the validation of the Lanthanum coverage was added to the Supplementary information.

1. The authors clarified the various effect of alkali promoters in the Introduction section, have the authors considered the effect of alkali on the acid-base properties of support.

Please note that we are not studying the effect of alkali promoters (La is a rare earth), alkali are mentioned in the introduction solely as an example of electropositive promoters, to which La belongs too. In the present study, we use unsupported Rh as model system and thus have not considered or discussed the promotor effect on the acid-base properties of support.

2. The authors build on in situ characterization. However, it should be noted that none of the characterization was conducted operando i.e while the reaction runs on the catalyst and gaseous products are thoroughly analyzed while the catalyst is being characterized. Can the authors provide the product distributions in addition to the FEM intensity?

Water is the sole product in H₂ oxidation and thus there is no particular product distribution. The product water was also detected by mass spectrometry, but in an averaging mode and not restricted to the nanotip. For this reason, the term “in situ” was used.

3. Can the authors provide the evidence for the relationship between the measured work function and La coverage? Is there formation of Rh-La bimetallic assemblies? Would the assemble effect or strain effect between Rh and La influence the work function?

The present study has been performed with submonolayer coverages of La prepared by room temperature evaporation and moderate annealing. It is well known from LEED studies that for such coverages wide-mesh La-structures are formed on transition metal surfaces (without forming La-substrate bimetallic ensembles), due to repulsive dipole-dipole interaction (see e.g. Refs. 21, 40, 41, 42 in the revised manuscript). The relationship between the measured work function and the La coverage including details of the La-coverage calibration (we thank the Reviewer for the suggestion) have now been added to the Supplementary information (Supplementary Note 1 and Supplementary Figure 1). We furthermore added Refs 21 and 41 to the revised manuscript and Refs. 1-21 to the Supplementary Information.

4. Is the reaction sensitive to O₂ pressure and coverage, i.e., would the reaction mechanism change if the H₂-O₂ reaction was performed at the high pressure ($p_{O_2} = 7.7 \times 10^{-7}$ mbar) and coverage of O₂ or the very low coverage of O₂?

The coverage of atomic oxygen (O, i.e. molecular oxygen, is not present on Rh under the given conditions) during the reaction does not depend on the O₂ pressure, but rather on the ratio between the H₂ and O₂ partial pressures. Furthermore, the reaction mechanism of H₂ oxidation on platinum group metals does not depend on the total pressure, at least not up to atmospheric pressure.

5. Would the H₂ be adsorbed on the surface of La? Previous report revealed the stabilization effect of H species by alkali ions (Nature Catalysis volume 3, pages703–709 (2020)).

We note that we are entering an absolute new territory with the present study: not only were H₂ adsorption on La surfaces or a La influence on H₂ adsorption on Rh not studied so far, even La adsorption on Rh was, to our knowledge, not investigated yet. The extrapolation of the stabilization effect of alkali observed in catalytic hydrogenation on atomically dispersed Ru to the present study of non-alkali (La) promotion of a solid Rh surface seems to be too far-fetched.

6. How did the authors determine the position of τ_B for the La/Rh systems?

The transition point τ_B is the inflection point on the local FEM intensity versus p_{H_2} as explained in Supplementary Note 2. We improved the explanation in Supplementary Note 2 and refer to Supplementary Note 2 in the revised manuscript on page 4.

7. There are many typos and edit errors for the references. Please carefully check them out.

We thank the Reviewer for this remark and have corrected the typos.

Reviewer #2

This work investigated the effects of co-adsorbed La in Rh nanotip on the H₂ oxidation with in-situ monitoring of FEM. In-situ characterization of catalytic reaction and structural evolution is of great interests and poses significant difficulties in both experimental and theoretical approaches. The authors identified different pacemakers between La-mediated Rh surface and La-free Rh surfaces and revealed the role of La using micro-kinetic modeling simulations. The

reviewer thinks the work is a good extension of their previous works such as 10.1021/acscatal.3c00060, 10.1038/s41467-021-26855-y, 10.1126/science.abf8107, on in-situ characterization of FEM for catalytic reactions and provides some new insights on the role of promoters in catalysis.

We thank the Reviewer for this encouraging comment.

Below are some minor comments for the further improvement of this work.

1. The authors investigated the La-induced phase changes. How could this finding differ from as-synthesized catalysts of La-Rh nanoparticles?

The general trends detected in the present study, namely shifts of kinetic transitions and change of pacemakers should also apply to La-Rh nanoparticles. The details, such as e.g. the extent of the shift, will, of course, depend on the size and crystallographic configuration of the particular particles. The La-content on/in a nanoparticle will also play a role.

2. The authors utilized microkinetic modeling to understand the experimental observations of different pacemakers. They adjusted the kinetic parameters of oxygen adsorption to incorporate the effects of electropositive La. What would be the effects of species with different electronegativity? Atomistic simulation with first-principles calculations may elucidate the effects of binding affinity change and predict local surface atomic configurations. The effect of promoters other than alkali has only been studied to a limited extent, which was the motivation to examine a rare earth element in the present study. It is difficult to predict the effect of species with different electronegativity, thus, more studies with different types of promoters are necessary.

We agree with the Reviewer that first-principles calculations may provide in the future reliable predictions of the effect of binding affinity variations and even predict the role of local surface atomic configurations.

Reviewer #3

The manuscript reports the tuning of the bistable reaction states with a La-mediated local catalytic effect. The experimental measurements are based on in-situ FEM imaging, from which the variation of the image contrast is used to extract the reaction kinetics and catalytic states of the catalyst surface. The authors provide convincing experimental results to show the effect of the La deposition on changing the bistability pattern of the reaction, and the fundamental insights are of interest to the catalysis community.

We thank the Reviewer for this encouraging comment.

The authors are suggested to address the following main issues:

1. introduction: some information about why the Rh/La system is of particular interest and importance would be necessary.

Lanthanum is a representative of the rare earth metals, which are characterized by a generally higher thermostability compared to alkali promoters adsorbed on transition metal surfaces. We have added this point to the introduction (page 2).

2. sample preparation: how the atomically clean Rh tip is ensured, particularly on the different

facets? Is there any direct evidence such as by XPS or AES to confirm the surface cleanliness? How are the three facets ((001), (011) and (111)) of the Rh tip ensured to have the same coverage of La from the evaporation deposition?

The atomic cleanliness of the Rh tip is ensured due to layer-by-layer field evaporation in ultrahigh vacuum: a procedure in which many surface layers are removed under FIM control with atomic resolution. The result is the ultimate evidence: each “foreign” atom would be directly visible in the FIM image. Due to the nm-size of the hemispherical tip sample, techniques such as XPS or AES are not applicable to be combined with FIM. Concerning the La coverage, the Rh sample was annealed after La evaporation to a temperature of 550 K at which surface diffusion of La equalizes the coverage. This was confirmed by FEM imaging. A statement regarding the La coverage equalization was added to the new Supplementary Note 1.

3. Fig. 2 shows the measurements in determining the p_{H_2} and T range between the active and inactive states. It would be helpful to show how many cycles of the cyclewise variations of p_{H_2} and T are performed to show the repeatability. Are there any differences in the hysteresis curves between different cycles, and does the catalyst (and promoter La) show any kind of deactivation that may induce changes in the hysteresis curves? The experiments are performed at elevated temperatures that may result in significant surface diffusion of La adatoms, for which the surface coverage of La may change over time. Any results and/or discussion of this dynamic nature of the catalyst surface facets on the evolution of the hysteresis curves would be interesting.

We agree with Reviewer that the evolution of the catalyst itself is an interesting topic. However, the present studies focused on La-promotor effects, thus precautions were taken to avoid such evolution. The necessity to avoid surface contamination by residual gases limited the duration of each experiment to few (4-5) cycles of parameter variations. Within this time, the repeatability was within the experimental error of the pressure gauge. Checking the work function after each experiment showed that the La-coverage on the imaged surface had not remarkably changed. The hours-long stability of self-sustained periodic oscillations at constant external parameters additionally confirms the constancy of the La-coverage. We mentioned this point in Supplementary Note 1.

4. Fig. 2 and 3 show that all the three facets ((001), (011) and (111)) have the synchronized oscillations between the active and inactive states (in terms of p_{H_2} and T), is this an artifact from the experiment measurements (imaging intensity) due to the close proximity of the three facets or an intrinsic behavior? If it is the latter, what is the specific reason for inducing the synchronized oscillations for the three facets with different surface structures and even different La coverage? How do the bi-stable reaction states (or self-organizing dissipative structures) depend on the local crystallography and La coverage of different facets?

The synchronization of the oscillations is not an artefact because it results from the diffusional coupling via adsorbed highly mobile hydrogen, as comprehensively discussed in our previous studies (Refs. 32, 33 in the revised manuscript). Please note that also unsynchronized oscillations can also be observed on nanotips (Ref. 31, 34 in the revised manuscript). We added this point to the revised manuscript on page 8.

The role of the local crystallography in the reaction states has been discussed in Refs. 61 and 62 of the revised manuscript. Although it is not possible to prepare different La coverages on adjacent nanofacets due to the surface diffusion of La, we intend to vary the La coverage in the entire reaction area in future studies.

5. Fig. 2c shows the shift of the bistability region to the higher p_{H_2} in the La/Rh system. This shift is attributed to the higher oxygen energy due to the La promoter effect. If this is true, the bistability region may also depend on the surface coverage of La, any experimental measurements of the bistability region for different La coverages would make a more convincing case.

The shift of the bistability region and the increase of the binding energy of oxygen adatoms due to electropositive coadsorbates are well-proven experimental facts. In addition, a similar shift was observed and explained for the CO oxidation reaction (see e.g. Ref. 20 and references therein). The present study is the first observation of this effect for H_2 oxidation. The role of varying La coverages will be targeted in future studies.

6. For the measurements shown in Fig. 3, what is the surface coverage of La for the (001), (110) and (111) facets? Do the primary peaks depend on La coverage? And why do the three facets show the same oscillation frequency? This seems to contradict with the statement “the oscillation frequency is very sensitive to local surface crystallography” on page 6.

The La-coverage for the (001), (110) and (111) facet was 0.2 ML, as mentioned throughout the manuscript, due to diffusion mediated equalization. The three facets show the same oscillation frequency due to diffusional coupling by hydrogen, as explained above. This does not contradict the above statement, as, when the diffusional coupling collapses, the oscillation frequency indeed varies with surface structure, as observed on planar samples (Ref. 60) and on nanofacets (Refs. 31, 34). We mentioned this on page 8 of the revised manuscript.

7. It is proposed in Fig. 4 and in the discussion section that the S-type pacemakers can be modified by the La coverage. Any results from experiment results with different La coverages would strengthen this point. It is proposed in Fig. 4e that the two types of the pacemakers are related to the high-index facets that may be unstable and evolve over the time at the elevated temperature. Are La also deposited onto these high index facets and how stable of the La adatoms on these high-index facets at the elevated temperature?

The La was deposited onto the whole hemispherical nanotip surface and the submonolayer La-coverage was indeed very stable at the present reaction temperatures: this the advantage of rare earth promoters, whose binding energy on transition metals is higher than that of alkali coadsorbates.

8. The measurements are performed at the extremely low pressure regime ($\sim 10^{-6}$ mbar), are the results relevant to the realistic catalytic reactions at a much high pressure? Is there a pressure gap regarding the effect of the La promoter on tuning the bi-stable reaction states?

The pressure gap is a general problem in catalysis, as any reaction exhibits it to a smaller or bigger extent. For H_2 oxidation, however, the overall reaction mechanism remains the same, unless the surface oxidation state changes. Electronic effects increasing the binding energy of

oxygen and thus shifting the kinetic transition points, as observed in the present study, should definitely also be detectable at higher pressures. The question is, of course, the extent of the effect. The present first observations enabled the detection and fundamental explanation of novel effects. Future studies (higher pressures, La-coverage variation, nanoparticles, etc.) will be directed toward understanding the role of reaction parameters, thus revealing possible pressure and materials gaps.

REVIEWER COMMENTS

Reviewer #1 (Remarks to the Author):

The authors have revised the manuscript according to the comments raised by the reviewers. However, some of the suggestions seems to be partially ignored by the authors, for example, Reviewer 1-4 (no evidence), Reviewer 2-2 (no additional DFT calculations). Particularly, I agree with reviewer 2 that DFT calculations seems to be essential to support their hypothesis. Therefore, I would suggest further revision before its publication.

Reviewer #3 (Remarks to the Author):

The authors have addressed my concerns, and I suggest for acceptance.

REPLY TO REVIEWER COMMENTS

Reviewer #1 (Remarks to the Author):

The authors have revised the manuscript according to the comments raised by the reviewers.

We thank the Reviewer for pointing this out.

However, some of the suggestions seems to be partially ignored by the authors, for example, Reviewer 1-4 (no evidence), Reviewer 2-2 (no additional DFT calculations). Particularly, I agree with reviewer 2 that DFT calculations seems to be essential to support their hypothesis. Therefore, I would suggest further revision before its publication.

We have now responded more explicitly to the comments:

Concerning Point 1-4:

4. Is the reaction sensitive to O₂ pressure and coverage, i.e., would the reaction mechanism change if the H₂-O₂ reaction was performed at the high pressure ($p_{O_2} = 7.7 \times 10^{-7}$ mbar) and coverage of O₂ or the very low coverage of O₂?

We have now added two references experimentally validating the Langmuir-Hinshelwood kinetics for hydrogen oxidation on (metallic) Rh, the first in the 10^{-4} mbar pressure range [Surf. Sci 254, 1991, 125] and the second demonstrating the reaction mechanism even for ambient pressure [Chem. Eng. Sci. 2013, 89, 171]. We have added a short comment to the main text in the discussion section (p. 14) stating that the reaction mechanism is generally accepted from high vacuum to atmospheric pressure. At very low oxygen coverage, the reaction is in its active steady state, since a sufficient number of surface sites is available for hydrogen to adsorb and react. Concerning higher pressure and excess of O₂, the reaction would rapidly be oxygen-poisoned.

Concerning Point 2-2:

2. The authors utilized microkinetic modeling to understand the experimental observations of different pacemakers. They adjusted the kinetic parameters of oxygen adsorption to incorporate the effects of electropositive La. What would be the effects of species with different electronegativity? Atomistic simulation with first-principles calculations may elucidate the effects of binding affinity change and predict local surface atomic configurations.

We have now carried out initial exploratory DFT calculations of the La-Rh system. The results show that the oxygen binding energy is indeed higher at La-Rh sites, corroborating the experimental observations and rationalizing the shift in the kinetic phase diagrams. We have added these new results as Supplementary Note 5 and mention them briefly in the main text on page 12. We have also made a stability analysis of the micro-kinetic model, commented on page S10 of the Supplementary Information, that illustrated its robustness in describing the La-modified oscillation behavior.

Reviewer #3 (Remarks to the Author):

The authors have addressed my concerns, and I suggest for acceptance.

We thank the Reviewer for agreeing with our answers/revisions.

REVIEWERS' COMMENTS

Reviewer #1 (Remarks to the Author):

Accept!